# The Current Treatment Trends and Survival Patterns in Melanoma Patients with Positive Sentinel Lymph Node Biopsy (SLNB): A Multicenter Nationwide Study

**DOI:** 10.3390/cancers15102667

**Published:** 2023-05-09

**Authors:** Marcin Ziętek, Paweł Teterycz, Jędrzej Wierzbicki, Michał Jankowski, Manuela Las-Jankowska, Wojciech Zegarski, Janusz Piekarski, Dariusz Nejc, Kamil Drucis, Bożena Cybulska-Stopa, Wojciech Łobaziewicz, Katarzyna Galwas, Grażyna Kamińska-Winciorek, Marcin Zdzienicki, Tatsiana Sryukina, Anna Ziobro, Agnieszka Kluz, Anna M. Czarnecka, Piotr Rutkowski

**Affiliations:** 1Department of Oncology, Wroclaw Medical University, 50-367 Wroclaw, Poland; 2Department of Surgical Oncology, Lower Silesian Oncology, Pulmonology and Hematology Center, 53-413 Wroclaw, Poland; jedrzejwierzbicki@gmail.co; 3Department of Soft Tissue/Bone Sarcoma and Melanoma, Maria Sklodowska-Curie National Research Institute of Oncology, 02-781 Warsaw, Polandmarcin.zdzienicki@pib-nio.pl (M.Z.); aannaziobro@gmail.com (A.Z.); anna.czarnecka@gmail.com (A.M.C.); piotr.rutkowski@pib-nio.pl (P.R.); 4Department of Computational Oncology, Maria Sklodowska-Curie National Research Institute of Oncology, 02-781 Warsaw, Poland; 5Laboratory of Immunopathology, Department of Experimental Therapy, Hirszfeld Institute of Immunology & Experimental Therapy, Polish Academy of Sciences, 53-114 Wroclaw, Poland; 6Chair of Surgical Oncology, Collegium Medicum Nicolaus Copernicus University, Oncology Center—Prof Franciszek Łukaszczyk Memorial Hospital, 85-796 Bydgoszcz, Polandzegarskiw@co.bydgoszcz.pl (W.Z.); 7Department of Surgical Oncology, Medical University of Lodz, 90-419 Lodz, Poland; janusz.piekarski@umed.lodz.pl (J.P.); dariusz.nejc@umed.lodz.pl (D.N.); 8Nicolaus Copernicus Multidisciplinary Center for Oncology and Traumatology, 93-513 Lodz, Poland; 9Department of Surgical Oncology, Gdansk Medical University, 80-210 Gdansk, Poland; 10Department of Clinical Oncology, Maria Sklodowska-Curie National Research Institute of Oncology, Cracow Branch, 31-115 Cracow, Poland; bcybulskastopa@vp.pl; 11Department of Surgical Oncology, Maria Sklodowska-Curie National Research Institute of Oncology, Cracow Branch, 31-115 Cracow, Poland; wlobaziewicz@wp.pl; 122nd Department of Radiotherapy and Chemotherapy, Maria Sklodowska-Curie National Research Institute of Oncology, Gliwice Branch, 44-102 Gliwice, Poland; 13Department of Bone Marrow Transplantation and Onco-Hematology, Skin Cancer and Melanoma Team, Maria Sklodowska-Curie National Research Institute of Oncology, Gliwice Branch, 44-102 Gliwice, Poland; dermatolog.pl@gmail.com; 14Faculty of Medicine, Medical University of Warsaw, 02-091 Warsaw, Poland; 15Department of Experimental Pharmacology, Mossakowski Medical Research Centers, Polish Academy of Sciences, 02-106 Warsaw, Poland

**Keywords:** melanoma, sentinel lymph nodes biopsy, completion lymph node dissection, active surveillance, adjuvant systemic treatment

## Abstract

**Simple Summary:**

In this study, the treatment trends and survival among 557 patients with sentinel lymph node biopsy (SLNB)-positive melanomas were analyzed. We have demonstrated the increasing role of the adjuvant systemic treatment and the non-proportional character in the RFS improvement during and after the adjuvant. The completion lymph node dissection (CLND) has, for years, been the standard of care for patients with clinically occult node-positive melanoma, although recently published multicenter randomized studies indicate a similar survival benefit for active surveillance in the groups where the multiple adjuvant systemic therapies have been implemented in patients after surgical resection of sentinel node metastases and in patients qualified for systemic adjuvant therapy without CLND. The limitation of our study was non-complete pathological reports outside reference oncological centers, especially in terms of the subtype of primary melanoma and the maximal size of the metastatic focus in the sentinel lymph node. Treatment of SLNB-positive melanoma patients is constantly evolving, and the role of surgery is currently rather limited. Whether CLND has been performed or not, in a group of SLNB-positive patients, adjuvant systemic treatment should be offered to all eligible patients.

**Abstract:**

Background: In melanoma treatment, an approach following positive sentinel lymph node biopsy (SLNB) has been recently deescalated from completion lymph node dissection (CLND) to active surveillance based on phase III trials data. In this study, we aim to evaluate treatment strategies in SLNB-positive melanoma patients in real-world practice. Methods: Five-hundred-fifty-seven melanoma SLNB-positive patients from seven comprehensive cancer centers treated between 2017 and 2021 were included. Kaplan–Meier methods and the Cox Proportional-Hazards Model were used for analysis. Results: The median follow-up was 25 months. Between 2017 and 2021, the percentage of patients undergoing CLND decreased (88–41%), while the use of adjuvant treatment increased (11–51%). The 3-year OS and RFS rates were 77.9% and 59.6%, respectively. Adjuvant therapy prolonged RFS (HR:0.69, *p =* 0.036)), but CLND did not (HR:1.22, *p =* 0.272). There were no statistically significant differences in OS for either adjuvant systemic treatment or CLND. Lower progression risk was also found, and time-dependent hazard ratios estimation in patients treated with systemic adjuvant therapy was confirmed (HR:0.20, *p =* 0.002 for BRAF inhibitors and HR:0.50, *p =* 0.015 for anti-PD-1 inhibitors). Conclusions: Treatment of SLNB-positive melanoma patients is constantly evolving, and the role of surgery is currently rather limited. Whether CLND has been performed or not, in a group of SLNB-positive patients, adjuvant systemic treatment should be offered to all eligible patients.

## 1. Introduction

The treatment of melanoma patients with sentinel lymph node metastases has changed over the past decade, and the recommended surgical management is now less invasive than it was previously [1,2]. The completion lymph node dissection (CLND) has, for years, been the standard of care for patients with clinically occult node-positive melanoma, although recently published multicenter randomized studies indicate a similar survival benefit for active surveillance in the groups where the multiple adjuvant systemic therapies have been implemented in patients after surgical resection of sentinel node metastases and in patients qualified for systemic adjuvant therapy without CLND [3,4].

The first Multicenter Selective Lymphadenectomy Trial (MSLT-1), announced in 2005, has proven the significance of sentinel lymph node biopsy (SLNB) in melanoma patients [5]. However, after 10-years follow-up, no statistically different melanoma-specific survival (MSS) rate was found in comparison between postponed CLND and immediate CLND at the time of clinically evident pathological nodes [6]. In a subsequent prospective randomized MSLT-2 trial, MSS also did not differ significantly between patients treated with CLND and those who had CLND performed only in the case of the clinical nodal involvement later on [7]. The German Dermatologic Cooperative Oncology Group (DeCOG) indicated conclusions and results of the multicenter randomized phase 3 trial that are important regarding this topic [8]. This study showed no differences in the distant metastasis-free survival (DMFS), recurrence-free survival (RFS) and overall survival (OS) between patients after CLND and observation after only 3 years of follow-up [8]. Similar conclusions—and, most importantly, no differences in survival—were realized after updating the same study following another 72 months of observation [9].

The results of these landmark studies have changed clinical practice, and CLND is no longer the standard of treatment in all melanoma patients with involved lymph nodes without distant metastases (stage III). These changes coincided with the increasingly important role of adjuvant systemic therapy in this group and an introduction of a wide pool of novel regimens [10,11,12,13]. Currently, the most common regimens of adjuvant therapy are based on a combination of dabrafenib (BRAF inhibitor) and trametinib (MEK inhibitor) in patients with *BRAF V600* mutation, as well as nivolumab or pembrolizumab (targeting programmed death receptor 1) in melanoma patients with or without *BRAF V600* mutation [14,15,16,17,18,19]. Systemic postoperative therapy has now become a standard treatment in clinical practice for high-risk patients after a radical resection of metastatic regional lymph nodes based on the results of clinical trials indicating a significant decrease in the risk of relapse.

The aim of this study was to retrospectively analyze the clinical outcome, prognostic factors and changes in surgical treatment in a group of melanoma patients with positive SLNB (stage III) in the context of recently published studies and the growing importance of adjuvant systemic treatment.

## 2. Materials and Methods

In total, 557 patients with active nodal disease confirmed by the SLNB, treated in 7 Polish comprehensive cancer centers between 1 December 2017 and 31 December 2021, were included in this study, and all of them were Caucasian. The cut-off for data was 30 June 2022. The exclusion criteria were presence of in-transit or satellite metastases or clinically detectable lymph node metastases before SLNB or distant metastases at primary staging. Four subgroups of patients were analyzed: patients treated with SNLB and CLND (1), patients treated with SLNB and CLND and at least one cycle of adjuvant therapy (2), patients treated with adjuvant therapy after SLNB (3) and patients only under active surveillance after SLNB with no surgical or systemic treatment (including those who did not agree to CLND and did not receive adjuvant therapy) (4).

Kaplan–Meier analysis was used for overall survival (OS) and relapse-free survival (RFS) estimation. Due to violated proportional hazard assumption, the RFS was modeled with time-dependent hazard ratios (HRs) for systemic adjuvant therapy (i.e., different HR during first year—“on treatment” and later—“subsequently”). The median follow-up was calculated using the reverse Kaplan–Meier method, where deaths were censored and the end of observation was treated as an event. Four baseline quantitative (age, melanoma primary tumor thickness, SLN positive count and SLN metastasis diameter) and eleven categorical variables (i.e., primary tumor histologic parameters, surgery site, surgical and systemic treatment history) were included in univariate analysis. Subsequently, Cox Proportional-Hazards Model was used in multivariate statistical analysis. For the multivariable Cox models, the variables with *p <* 0.10 in the univariable models were selected. Additionally, variables related to adjuvant systemic therapy and CLND were prespecified in both OS and DFS models. All analyses were performed using R 4.2.2 (R Foundation for Statistical Computing, Vienna, Austria).

The study was conducted In accordance with the principles of the Declaration of Helsinki. Informed consent was obtained from all subjects and/or their legal guardian(s). Ethical approval was provided by Bioethical Committee at Maria Sklodowska-Curie National Research Institute of Oncology (protocol code 3/2012 and date of approval: 18 December 2012).

## 3. Results

### 3.1. Clinical Characteristics of the Study Group

The study included 312 men (56.0%) and 245 women (44.0%), and the mean age was 58.0 years ± 15.9 years (Table 1). There were 139 patients (25.0%) with T1 or T2 stage, 179 patients (32.1%) with T3 and 239 patients (42.9%) with T4 at the diagnosis. In 194 (34.8%) and 107 patients (19.2%), nodular melanoma (NM) and superficial spreading melanoma (SSM) histologic subtypes were diagnosed, respectively. In the rest of the study group (n = 256; 46.0%), the subtype was other or unspecified. Ulceration was recognized in 343 patients (61.6%) and positive BRAF status in 278 patients (49.9%). The site of surgery of primary tumors was head and neck in 40 patients (7.2%); upper limb in 100 patients (18.0%); trunk in 251 patients (45.1%); lower limb in 129 patients (23.2%) and indefinite (no data provided) site in 37 patients (6.6%). The average number of positive SLN was 1.2 ± 0.7, and the number of totally resected SLN was 2.5 ± 2.0. In 112 patients (20.1%), nodal metastasis was less than or equal to 1 mm, in 333 patients (59.8%) over 1 mm and in another 112 patients (20.1%) unknown.

Out of 557 patients with SLNB-positive melanoma included in the study, 248 patients (44.5%) were treated with CLND without subsequent systemic therapy in an adjuvant setting. Further, 116 patients (20.8%) were treated with CLND and adjuvant systemic therapy. In 79 patients (14.2%), adjuvant systemic therapy was the only treatment after SLNB. In total, 364 patients (65.4%) underwent CLND, and 195 (35%) underwent at least one cycle of adjuvant systemic therapy. BRAF/MEK inhibitors and PD-1 inhibitors were administered in 79 patients and 115 patients, respectively. In 114 patients (20.5%), active surveillance strategy was used (ultrasound of regional nodes instead of CLND and no systemic treatment). Median follow-up time, from the start of treatment to death or end of observation, was 25.0 months (95% CI: 23.1 months–27.7 months).

### 3.2. Treatment Trends

Changes in the treatment trends, broken down into years, were shown in Figure 1. During the study course, a slight increase in total number of SLNB-positive melanoma patients treated per year was reported (Figure 1A). Between 2017 and 2021, significant changes in the treatment approach, including surgical and conservative treatment, were observed. There was a decrease in percentage of patients undergoing CLND only (78.9–16.7%) and an increase in the adjuvant treatment (1.1–31.2%) (Figure 1B). The percentage of patients after combined therapy (CLND + adjuvant therapy) also increased from 9.5% to 23.9%. The percentage of patients under active surveillance did not fluctuate significantly in the years 2018–2020; however, an increase was observed over the course of the analysis (10.5–28.3%).

### 3.3. Relapse-Free Survival Analysis

In the whole group, the 3-years RFS was 59.6% (95% CI: 54.5–65.2%). Within the subgroup treated with CLND, it was 68.5%, and 69.0% in those patients who did not undergo CLND (Figure 2A). The 3-years RFS was 57.1% in the group not treated with adjuvant systemic therapy, 71.1% in the group treated with BRAF-MEK inhibitors regiment and 56.7% in those treated with PD1 inhibitors (Figure 2B). Overall, relapse was noted in 148 cases. Further, 69 (47% of relapses) patients were diagnosed with distant metastases only; 37 (25% of relapses) were only diagnosed with new locoregional lesions and 42 (28% of relapses) patients experienced simultaneous recurrence in the form of both distant metastases and regional relapse.

Breslow thickness, T stage, presence of ulceration, BRAF mutation, histologic subtype, SLN metastasis diameter (quantitative), adjuvant treatment, systemic treatment regimen and treatment group were identified as negative prognostic factors in the univariate analysis for RFS. Significant correlation with poor prognosis was confirmed for T4 or unknown stage (*p =* 0.012), unspecified and other histologic subtype of primary melanoma (*p =* 0.036) and the systemic treatment regimen in the multivariate model (Table 2). However, systemic treatment regimen was a significant factor only taking into consideration different HR during the first year (“on treatment” groups). In this model, there were no statistically significant differences in RFS for CLND (*p =* 0.871).

### 3.4. Overall Survival Analysis

The 3-year OS rate was 77.9% (95% CI: 73.5–82.7%) in the whole group. In patients after CLND and without performed CLND, it was 77.5% and 81.2%, respectively (Figure 3A). The 3-year OS rates based on adjuvant systemic therapy grouping were as follows: no adjuvant 76.3%, BRAF-MEK inhibitors 76.6% and PD1 inhibitors 80.3% (Figure 3B).

The univariate analysis of negative prognostic factors for OS identified age, Breslow thickness, primary tumor stage, presence of ulceration, BRAF mutation, histologic subtype and SLN metastatic maximal diameter as significant. There were no statistically significant differences in OS for either CLND (*p =* 0.801) or adjuvant systemic treatment (*p =* 0.187). Regarding multivariate analysis, age (*p <* 0.001), presence of melanoma ulceration (*p =* 0.006), unspecified and other histologic subtype of melanoma (*p =* 0.015) and positive BRAF mutation (*p =* 0.040) were correlated with poor OS (Table 3).

## 4. Discussion

Although survival rates have been increasing in the SLNB-positive patients in recent years, they are still unsatisfactory in Poland and lower than European study results [20,21]. The study group in our analysis consisted of a majority of patients with advanced primary tumor thickness (i.e., 43% of patients were T4), and it differed from the MLST and DECOG populations, where, primarily, the patients had thin or intermediate-thickness melanoma (below 3.5 mm) [4,5,6,7,8,22]. This is still due to the relatively late diagnosis in Polish patients, which is reflected in the high average of melanoma tumor thickness at diagnosis (approximately 2 mm in Poland versus below 1 mm in Western Europe) [23]. Nonetheless, we have observed similar results and confirmed irrelevant differences between CLND and active surveillance. The reason for the small survival improvement in the adjuvant-treated group may be twofold; however, consistent with the literature data, the following factors should be considered: firstly, high risk of bias in this specific group and study [5,24]; secondly, overly short follow-up [25,26]. Nonetheless, we have observed improvement in RFS after adjuvant therapy, especially in the BRAF/MEK-inhibitor-treated group, which is consistent with data from clinical trials demonstrating that, in BRAF-mutated stage III patients, the impact of adjuvant therapy with BRAF/MEK for preventing melanoma relapses is higher during the first year of active therapy than in patients treated with immunotherapy. The RFS curves for patients treated with targeted therapy and anti-PD-1 agents overlap after approximately 2 years, so clear criteria are lacking for choosing one of these therapies in the adjuvant setting [14,15,17,27].

In this research, the real-world treatment trends in melanoma patients with involved lymph nodes were investigated in a population-wide study. Three recently published landmark studies have remarkably influenced the surgical management of SLNB-positive melanoma patients [20,26,28]. Within our nationwide study group in 2017, CLND was performed in the vast majority of patients (88%) and decreased to 41% in 2021. On the contrary, the number of patients treated with adjuvant systemic therapy increased from 11% to 51% in this period of time. The initial small number of patients is due to the fact that, before 2018, this treatment was available in Poland only in clinical trials. Furthermore, before the reimbursement of treatment in January 2021, it was accessible for patients only based on individual case applications for governmental Emergency Access to Drug Technologies. The watch and waiting strategy and active surveillance of patients with small (less than 1 mm) and single metastasis to SLN and no other risk factors is gaining importance, which was reflected in an increase in the percentage of patients from 11% to almost 30% within the study group. This trend may be related to the growing group of patients with less than 1 mm nodal metastasis as a result of earlier detection of regionally advanced melanoma [18].

The limitation of our study was non-complete pathological reports outside reference oncological centers, especially in terms of the subtype of primary melanoma and the maximal size of the metastatic focus in the sentinel lymph node. Nevertheless, the results of our study showed that subtype of melanoma other than NM/SSM or unspecified had prognostic value, which is in line with other reports [29].

The group of patients with SLNB-positive melanoma requires further analysis, and, in view of recently published studies and articles, prolonged follow-up is needed to state whether less surgery but more systemic treatment will improve the outcomes in SLN-positive melanoma patients. Although clinical data obtained after CLND provide valuable information regarding the disease advancement, patient survival and prognosis remain at similar levels to those when adhering to the active surveillance [30]. Due to the fact that CLND is associated with a high risk of complications and an increase in adverse events, limiting this approach to specific groups of patients and implementing it only when indicated will reduce the number of unnecessary operations [31]. We have observed changes in trends of surgical approach in SLNB-positive patients in Poland after introduction of Polish national guidelines [32,33,34], leading to eliminating CLND as well as a rapid increase in the use of systemic therapy in this population upon it becoming available. In accordance with those guidelines, the CLND might be performed in patients with a very high risk of extra-sentinel lymph node metastases, such as a large sentinel lymph node metastasis (larger than 1 mm metastasis), involvement of more than two sentinel lymph nodes with metastases or sentinel lymph node extracapsular infiltration. The active observation option remains a valid approach only in patients with small (up to 1 mm) metastatic deposits in SLN.

## 5. Conclusions

The role of surgery in SLNB-positive patients seems to be diminishing, and the CLND should be limited to some selected cases with high-risk features. We identified that the higher pT stage, unspecified and other histologic subtype of primary melanoma (including acral lentiginous melanoma) and the systemic treatment were independent prognostic factors for RFS, but patients’ age, presence of melanoma ulceration, unspecified and other histologic subtype of melanoma and positive BRAF mutation correlated with poorer OS. The systemic treatment decreases the risk of relapse in microscopic stage III disease and should always be offered in SLNB-positive patients regardless of whether CLND has been performed or not.

## Figures and Tables

**Figure 1 cancers-15-02667-f001:**
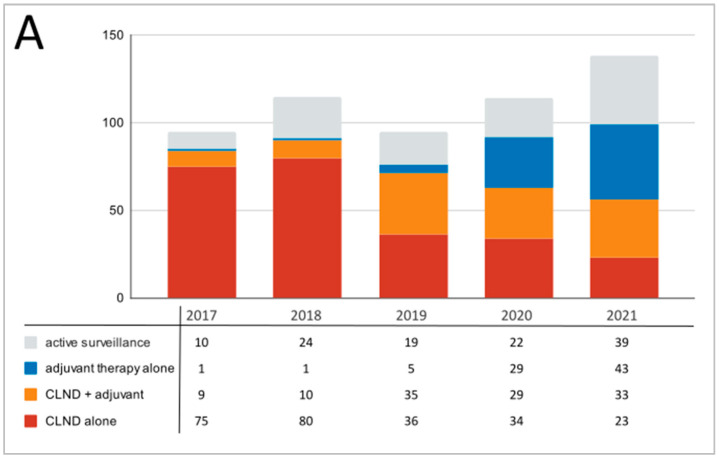
Observed trends in changes in management of melanoma patients’ therapy between 2017 and 2021; (**A**)—absolute numbers; (**B**)—percentage distribution.

**Figure 2 cancers-15-02667-f002:**
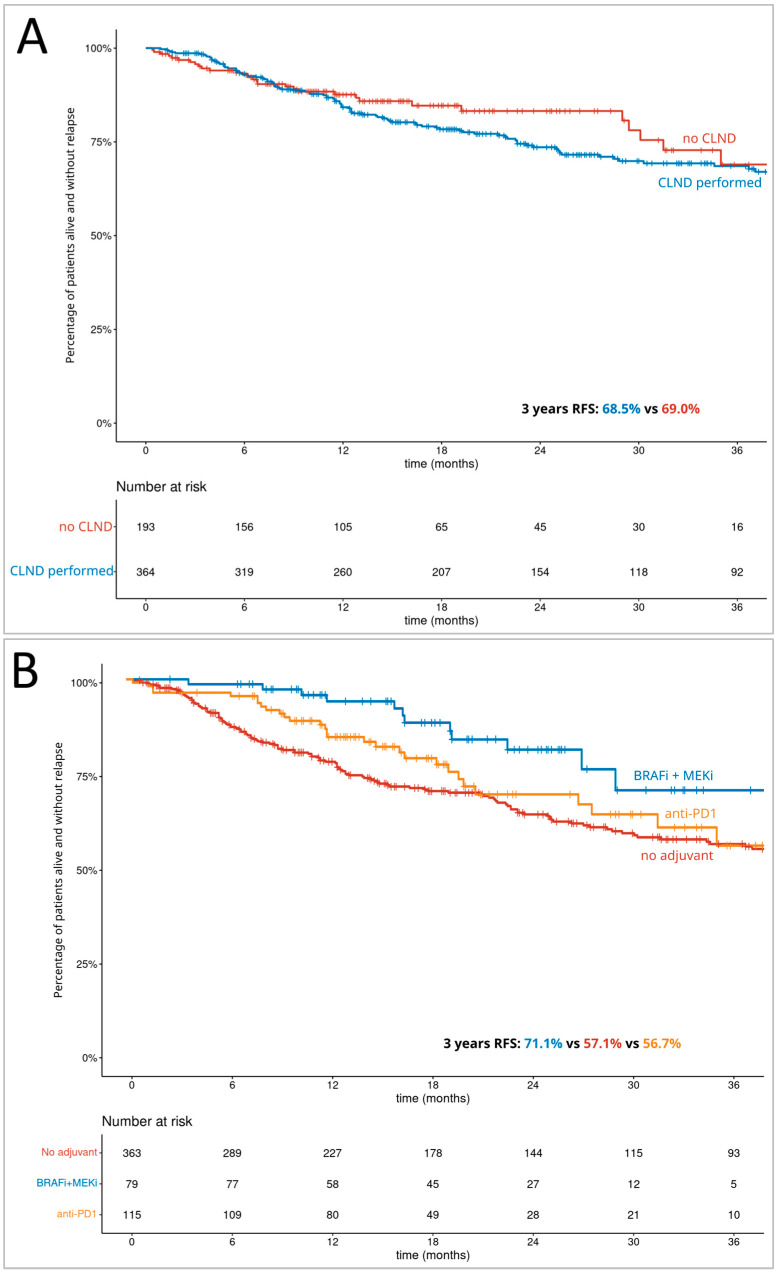
The Kaplan–Meier curves of the relapse-free survival for (**A**)—completion lymph node dissection; (**B**)—adjuvant systemic treatment.

**Figure 3 cancers-15-02667-f003:**
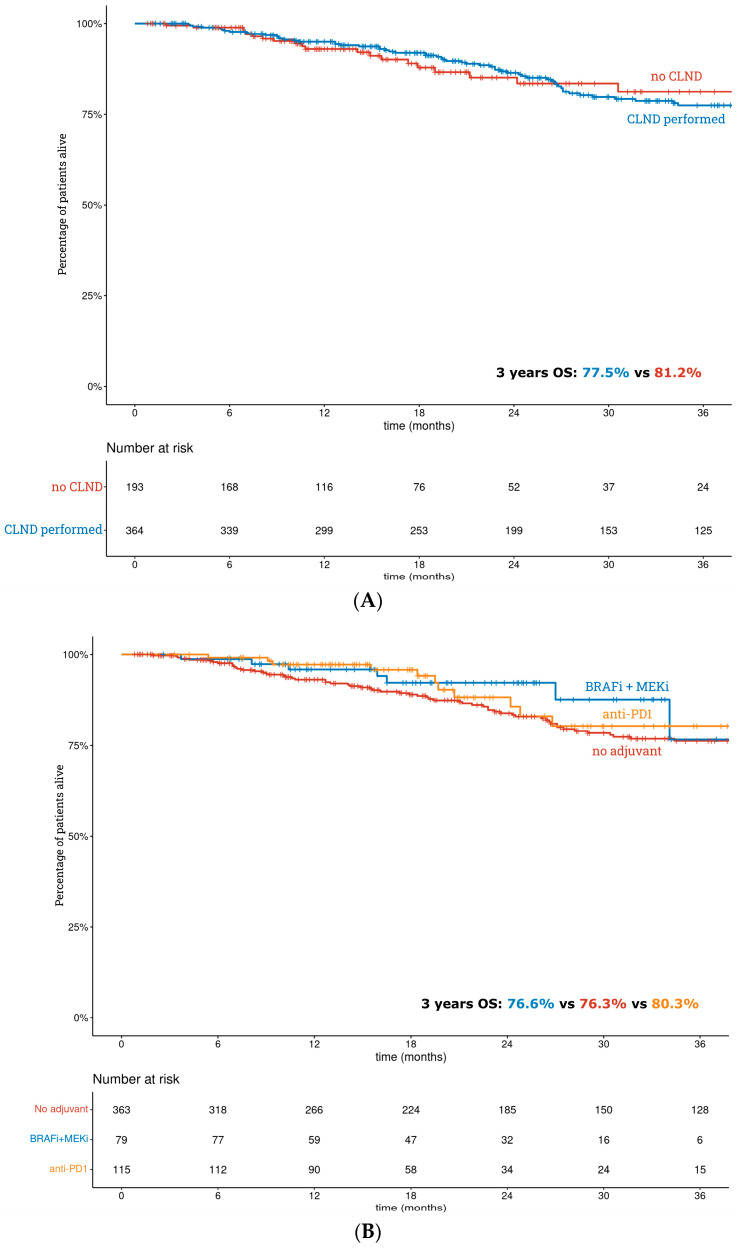
The Kaplan–Meier curves of the overall survival for (**A**)—completion lymph node dissection; (**B**)—adjuvant systemic treatment.

**Table 1 cancers-15-02667-t001:** Baseline characteristics of the SLNB-positive patients.

Factors		Number (n)	Percentage (%)
Treatment group	CLND ^1^ alone	248	44.5
	CLND ^1^ + adjuvant	116	20.8
	Adjuvant therapy alone	79	14.2
	Observation only after SLNB ^2^	114	20.5
Gender	male	312	56.0
	female	245	44.0
Age (mean ± SD, years)		58.0 ± 15.9	
T stage	T1–T2	139	25.0
	T3	179	32.1
	T4 or unknown	239	42.9
Melanoma histologic subtype	NM ^3^	194	34.8
	SSM ^4^	107	19.2
	unspecified and other	256	46.0
Melanoma ulceration	yes	343	61.6
	no	202	36.3
	unknown	12	2.2
BRAF status	positive	278	49.9
	negative	162	29.1
	unknown	117	21.0
Surgery site of primary tumor	head and neck	40	7.2
	upper limb	100	18.0
	trunk	251	45.1
	lower limb	129	23.2
	unknown	37	6.6
SLN ^5^ positive count (mean ± SD)		1.2 ± 0.7	
SLN ^5^ total number (mean ± SD)		2.5 ± 2.0	
SLN ^5^ metastasis diameter	less or equal 1 mm	112	20.1
	over 1 mm	333	59.8
	unknown	112	20.1
CLND ^1^ performed	yes	364	65.4
	no	193	34.6
Adjuvant therapy	yes	194	34.8
	no	363	65.2
Systemic treatment regimen	BRAFi/MEKi ^6^	79	14.2
	PD1 ^7^	115	20.6

1—completion lymph node dissection; 2—sentinel lymph node biopsy; 3—nodular melanoma; 4—superficial spreading melanoma; 5—sentinel lymph node; 6—BRAF inhibitors; MEK inhibitors (dabrafenib with trametinib); 7—programmed death 1 inhibitors (nivolumab or pembrolizumab).

**Table 2 cancers-15-02667-t002:** The univariate and multivariate analysis for relapse-free survival in patients with SLNB-positive melanoma.

Factors		Univariate Analysis		Multivariate Analysis	
		HR (95% CI)	*p*	HR (95% CI)	*p*
T stage (ref. level: T1/T2)	T3	1.43 (0.89–2.28)	0.136	1.12 (0.68–1.84)	0.652
	T4 or unknown	2.57 (1.68–3.93)	0.001	1.82 (1.13–2.92)	0.012
Melanoma histologic subtype (ref. level: = NM ^1^)	SSM ^2^	0.48 (0.30–0.77)	0.003	0.72 (0.44–1.20)	0.215
	unspecified and other	0.71 (0.51–0.97)	0.033	0.70 (0.50–0.98)	0.036
Melanoma ulceration (ref. level: no)	present	2.15 (1.51–3.05)	0.001	1.69 (1.14–2.50)	0.009
CLND ^3^ (ref. level: = no)	yes	1.22 (0.86–1.73)	0.272	1.03 (0.71–1.49)	0.871
Systemic treatment regimen (ref. level: = none)	BRAFi ^4^: group on treatment *	0.47 (0.27–0.84)	0.011	0.20 (0.07–0.56)	0.002
	BRAFi: group subsequently *			0.83 (0.39–1.78)	0.634
	PD1i ^5^: group on treatment *	0.83 (0.56–1.23)	0.357	0.50 (0.28–0.87)	0.015
	PD1i: group subsequently *			1.09 (0.61–1.95)	0.784

* Due to violated proportional hazard assumption, the RFS was modeled with time-dependent hazard ratios for systemic adjuvant therapy (i.e., different HR during first year—“on treatment” and later—“subsequently”). 1—nodular melanoma; 2—superficial spreading melanoma; 3—completion lymph node dissection; 4—BRAF inhibitors; MEK inhibitors (dabrafenib with trametinib); 5—programmed death 1 inhibitors (nivolumab or pembrolizumab).

**Table 3 cancers-15-02667-t003:** The univariate and multivariate analysis for overall survival in patients with SLNB-positive melanoma.

Factors		Univariate Analysis		Multivariate Analysis	
		HR (95% CI)	*p*	HR (95% CI)	*p*
Age (quantitative) (per 1 year change)		1.04 (1.02–1.05)	0.001	1.03 (1.01–1.05)	0.001
T stage (ref. level: = T1/T2)	T3	1.99 (1.01–3.92)	0.048	1.40 (0.69–2.84)	0.345
	T4 or unknown	2.81 (1.49–5.31)	0.001	1.54 (0.78–3.07)	0.218
Melanoma histologic subtype (ref. level: = NM ^1^)	SSM ^2^	0.42 (0.21–0.81)	0.010	0.61 (0.30–1.22)	0.163
	unspecified and other	0.46 (0.29–0.72)	0.001	0.55 (0.34–0.89)	0.015
Melanoma ulceration (ref. level: = no)	present	3.21 (1.81–5.70)	0.001	2.37 (1.28–4.37)	0.006
BRAF status (ref. level: = negative)	positive	0.44 (0.27–0.72)	0.001	0.56 (0.33–0.97)	0.040
	unknown			0.84 (0.47–1.55)	0.593
CLND ^3^ (ref. level: = no)	yes	0.94 (0.58–1.53)	0.801	0.91 (0.56–1.51)	0.739
Systemic treatment regimen (ref. level: = none)	BRAFi ^4^	0.70 (0.33–1.46)	0.343	1.16 (0.49–2.75)	0.741
	PD1 ^5^	0.74 (0.40–1.38)	0.342	0.82 (0.42–1.60)	0.562

1—nodular melanoma; 2—superficial spreading melanoma; 3—completion lymph node dissection; 4—BRAF inhibitors; MEK inhibitors (dabrafenib with trametinib); 5—programmed death 1 inhibitors (nivolumab or pembrolizumab).

## Data Availability

The data presented in this study are available uppon reasonable request from the corresponding author. The data are not publicly available due to legal and ethical reasons.

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
