# Peer review of "The Current Treatment Trends and Survival Patterns in Melanoma Patients with Positive Sentinel Lymph Node Biopsy (SLNB): A Multicenter Nationwide Study"

_cancers, 2023, doi:10.3390/cancers15102667_

Round 1

Reviewer 1 Report

Interesting and valuable work. Based on a retrospective multicenter study, the authors confirmed that the "watch and wait" approach in treating melanoma patients with metastases to lymph nodes after SLNB with favorable prognostic factors makes sense. Of particular value is the demonstration of the limited role of CLND in making decisions about adjuvant treatment. The possibilities of doing so were known. Here, the authors clearly and comprehensively demonstrated the advantages of such treatment for the patient's benefit (probability of survival). Reducing the number of CLNDs certainly also improves the quality of life of these patients.

Additionally, I have a few minor comments.

1. Results: in 256 (46%) patients, the subtype of melanoma was other or unspecified; in 112 (20.1%) patients, the size of the metastatic focus in the lymph node was unknown. These limitations should be emphasized in the discussion.

2.  avoid starting a sentence with a number 

3. 3.2.: more than twofold increase in supervision is difficult to describe as small (10.5-28.3%)

4.3.4. - better: the probability of overall survival - in Figures 2 and 3, respectively, a and b, survival data (RFS, OS) differ from those in the text. They should be thoroughly edited figures and explain what the difference is.

5. Discussion: I propose to describe the abbreviations MLST and DECOG (Multicenter Selective Lymphadenectomy Trial and German Dermatologic Cooperative Oncology Group)

6. Conclusions: There are no endpoints related to the previously set goal - prognostic factors

7. References: try to avoid self-citation, exceptionally justifying such a need.

Author Response

  1. Results: in 256 (46%) patients, the subtype of melanoma was other or unspecified; in 112 (20.1%) patients, the size of the metastatic focus in the lymph node was unknown. These limitations should be emphasized in the discussion. 

Thank you for this comment. These limitations are due to not completed histopathological reports outside reference oncological centers, what is related to retrospective nature of our study. The subtype of melanoma is not obligatory pathological description and not included in the staging system. It has been added to the discussion.

  1. avoid starting a sentence with a number  

Author response: Thank you for pointing this out. The reviewer is correct, and we have introduced some changes in the main text.

  1. In total, 557 patients with active nodal disease confirmed by the SLNB, treated in 7 Polish comprehensive cancer centers between 1 December 2017 and 31 December 2021, were included in this study, and all of them were Caucasian.

3.2. 79 patients and 115 patients were treated with BRAF/MEK inhibitors and PD-1 inhibitors, respectively.

This sentence has been changed to:

BRAF/MEK inhibitors and PD-1 inhibitors  were administered in 79 patients and 115 patients , respectively.

  1. 3.2.: more than twofold increase in supervision is difficult to describe as small (10.5-28.3%) 

Author response: Thank you for this valuable comment. We have made appropriate changes to the text.

  1. 2.:The percentage of patients under active surveillance did not fluctuate significantly, however, a slight increase was observed over the course of the analysis (10.5% - 28.3%).

This sentence has been changed to:

The percentage of patients under active surveillance did not fluctuate significantly in the years 2018 - 2020, however, an increase was observed over the course of the analysis (10.5% - 28.3%).

4.3.4. - better: the probability of overall survival - in Figures 2 and 3, respectively, a and b, survival data (RFS, OS) differ from those in the text. They should be thoroughly edited figures and explain what the difference is. 

Author response: We agree with the reviewer’s assessment. Accordingly, throughout the manuscript, we have revised such paragraphs. We have replaced the pictures with appropriate ones that are consistent with the text.

  1. Discussion: I propose to describe the abbreviations MLST and DECOG (Multicenter Selective Lymphadenectomy Trial and German Dermatologic Cooperative Oncology Group) 

Author response: Thank you for your valuable advice. However, the abbreviations have been explained in the Introduction chapter.

  1. Conclusions: There are no endpoints related to the previously set goal - prognostic factors 

Author response: The end-points of the study were: clinical outcome, prognostic factors and changes in surgical treatment in a group of melanoma patients with positive SLNB (stage III). We have added these information in conclusion part.

  1. References: try to avoid self-citation, exceptionally justifying such a need.

Author response: The citations used have a very important aspect in the presentation of the topic, especially if the material concerns Polish data from reference centers. It is therefore necessary to leave them in their current wording and numbering.

Reviewer 2 Report

Nicely designed study showing that potentially mutilating surgery does not improve survival, but adjuvant therapy does! 

Author Response

Author response: Thank you for your comments.

Reviewer 3 Report

The authors retrospectively analyze the clinical outcome, prognostic factors and changes in surgical treatment in a group of melanoma patients with positive SLNB (stage III)

This paper is very interesting since sentinel lymph node dissection is no longer the standard of treatment in all melanoma patients.

I have a few substantive questions/comments to consider:

·         It is recommended to add some sentences,  in the introduction, to further describe adjuvant systemic therapy

·         Moreover, in the discussion, the authors should extensively discuss their results with respect to what is reported in the literature on the use of adjuvant therapies

Minor editing

Author Response

Author response: Thank you for this remark, we have added sentences to the discussion and in the introduction.

Reviewer 4 Report

It is an interesting study on the real-life situation of melanoma treatment, given the evolution to practice less completion lymph node dissection (CLND) and the current use of adjuvant therapies.

My comments on improvements to be made:

1.       To better understand the meaning of the word nationwide, can the authors specify what percentage of the population of Poland is treated in the 7 referred centers?

2.       The lower RFS at 3 years among the patients treated with IBRAF or PD1 inhibitors is striking. Are the criteria for the use of one strategy or another known in BRAF+ patients, who also have a worse prognosis? Authors should explain this point or consider it bias. It is true that the type of adjuvant treatment is not an independent prognostic factor for OS in the multivariate study. It can also be further discussed why BRAF is not an independent prognostic factor for RFS and it is for OS.

3.       T unknown is included within T4. Although I understand that if T is not known, it may be because they are locally advanced melanomas, the authors must justify the reason to analyze together with T4.

4.       I understand the criteria for performing CLND has been variable in each center, in time and depending on the physician. The authors can discuss the current indication of CLND.

5.       It is true that currently the nodal status assessment is limited as CLND is not systematically carried out. But I think it would have been relevant to include the distribution of stages in the characteristics of the patients and to correlate these with the use of adjuvant treatment and with the type of adjuvant treatment. Obviously, this should change the variables analyzed, since stage includes ulceration and Breslow. The stages according to TNM 8th version are currently used for decision making. Not having information according to stages is one of the major limitations of the study and should be discussed as a source of bias.

Mistake: when figure 2 is referred to in the text, it corresponds to figure 3 and vice versa. In addition, in the figure about RFS, is written OS inside.

Author Response

To better understand the meaning of the word nationwide, can the authors specify what percentage of the population of Poland is treated in the 7 referred centers?

Author response: We have included 7 largest Polish centers and according to data from Polish state insurance company we have covered about 65% of Polish population.

  1. The lower RFS at 3 years among the patients treated with IBRAF or PD1 inhibitors is striking. Are the criteria for the use of one strategy or another known in BRAF+ patients, who also have a worse prognosis? Authors should explain this point or consider it bias. It is true that the type of adjuvant treatment is not an independent prognostic factor for OS in the multivariate study. It can also be further discussed why BRAF is not an independent prognostic factor for RFS and it is for OS.

Author response: The lack of information about the BRAF mutation was related to the fact that it was not routinely tested in each patient. In the early years of the analysis, the adjuvant was not widely available in Poland, but only in clinical trials or on individual request (as described in the discussion), so not all stage III melanoma had the result of this test. Nowadays it is obligatory testing for all stage III and IV melanomas and recommended for stage IIC.

We have added these points about the difference in RFS to the discussion section:

It is consistent with data from clinical trial demonstrating that in BRAF-mutated stage III patients the impact of adjuvant therapy with BRAF/MEK for preventing melanoma re-lapses is higher during the first year of active therapy than in patients treated with immunotherapy. The RFS curves for patients treated with targeted therapy and anti-PD-1 agents overlap after approximately 2 years, so there are not clear criteria for choice one of these therapies in adjuvant setting.

On the other hand BRAF-mutated melanomas may behave more aggressively in situation of distant metastases what may have impact on OS.

  1. T unknown is included within T4. Although I understand that if T is not known, it may be because they are locally advanced melanomas, the authors must justify the reason to analyze together with T4.

Author response: T unknown was the category described when the Breslow was not achievable or not assessed. Unknown T stage was analyzed together with T4 due to the data-driven analysis - these two groups of patients had the same poor prognosis.

  1. I understand the criteria for performing CLND has been variable in each center, in time and depending on the physician. The authors can discuss the current indication of CLND.

Author response: Thank you for your valuable advice. We have added adequate sentence considering CLND indication, which is listed in section 4.3 in the main text of the manuscript.

In accordance with the Polish guidelines the CLND might be performed in patients with a very high risk of extra-sentinel lymph node metastases, such as a large sentinel lymph node metastasis (larger than 1mm metastasis), involvement of more than 2 sentinel lymph nodes with metastases, or sentinel lymph node extracapsular infiltration.

  1. It is true that currently the nodal status assessment is limited as CLND is not systematically carried out. But I think it would have been relevant to include the distribution of stages in the characteristics of the patients and to correlate these with the use of adjuvant treatment and with the type of adjuvant treatment. Obviously, this should change the variables analyzed, since stage includes ulceration and Breslow. The stages according to TNM 8th version are currently used for decision making. Not having information according to stages is one of the major limitations of the study and should be discussed as a source of bias.

Author response: Thank you for this comment. These limitations are due to not completed histopathological reports outside reference oncological centers, what is related to retrospective nature of our study. We have added this in the discussion section as a limitation of the study and a potential source of bias. While we respectfully agree with your opinion, as you suggested, adding the new correlations may change previously analyzed variables.

Reviewer 5 Report

This is a well designed and well described study.

There are just some minor points that might improve clarity for readers.

The concept of "violated proportional hazard assumption" needs a short explanation as to why and how this differs from the usual process.

It would be helpful to add P values to Figures 2 and 3, or to where these results appear in the text. 

The labelling of Figure 2: the legend (and text) refer to this as relapse-free survival but within the graphs themselves this appears as "3 years OS".

In the final part of the Discussion, you state that active observation only remains a valid option only for small SNB metastatic deposits. Would not age or frailty also be included in this, as there might be patients who could undergo SNB but not a prolonged course of adjuvant treatment? Is it also possible that age could be a confounder in the analyses, as older patients might be less likely to be selected for adjuvant therapy, thereby worsening the outcome for older patients and those not receiving adjuvant therapy? Perhaps an additional analysis might easily address this question?

Generally ok.

3.3 line 3 "undergo"

3.3 line 4: "regimen"

Author Response

The concept of "violated proportional hazard assumption" needs a short explanation as to why and how this differs from the usual process.

Author response: Thank you for pointing this out. To our knowledge, this is the first study to highlight the non-proportional character in the RFS improvement. We have shown that the hazard of relapse or death is greatly reduced with adjuvant treatment in the treatment period. After that the rate of relapse or death is similar between the groups. These results suggest that adjuvant treatment helps the most in patients who otherwise would relapse in a short time period and, simultaneously, its effect is not lost in further follow-up years. This observation of non-proportional benefit in hazard ratio may be important for the designing of the future clinical trials in the adjuvant setting.

It would be helpful to add P values to Figures 2 and 3, or to where these results appear in the text. 

Author response: While we appreciate the reviewer’s feedback, we respectfully disagree. The groups shown in figures 2 and 3 had different baseline characteristics and were not randomly assigned. Therefore no formal testing is allowed here. Omitting p values was a conscious decision. 

The labelling of Figure 2: the legend (and text) refer to this as relapse-free survival but within the graphs themselves this appears as "3 years OS".

Author response: We agree with the reviewer’s assessment. Accordingly, throughout the manuscript, we have revised such paragraphs. We have replaced the pictures with appropriate ones that are consistent with the text.

In the final part of the Discussion, you state that active observation only remains a valid option only for small SNB metastatic deposits. Would not age or frailty also be included in this, as there might be patients who could undergo SNB but not a prolonged course of adjuvant treatment? Is it also possible that age could be a confounder in the analyses, as older patients might be less likely to be selected for adjuvant therapy, thereby worsening the outcome for older patients and those not receiving adjuvant therapy? Perhaps an additional analysis might easily address this question?

Author response: Thank you for your very valuable advice. We think it would be a very interesting analysis for the next article concerning the possible analysis of SNB without prolonged course of adjuvant treatment in older patients.

Round 2

Reviewer 3 Report

The manuscript can be accepted